# Seroepidemiology and associated risk factors of hepatitis B and C virus infections among pregnant women attending maternity wards at two hospitals in Swabi, Khyber Pakhtunkhwa, Pakistan

**Muhammad Israr**[1,2]*, **Fawad Ali**[3], **Arif Nawaz**[3], **Muhammad Idrees**[4], **Aishma Khattak**[5], **Shafiq Ur Rehman**[1], **Azizullah Azizullah**[1], **Bashir Ahmad**[1], **Syeda Asma Bano**[6], **Rashid Iqbal**[7]

1 Department of Biology, The University of Haripur, Haripur, Khyber Pakhtunkhwa, Pakistan, 2 College of Life Science, Hebei Normal University, Shijiazhuang, Hebei, PR China, 3 Department of Chemistry, Bacha Khan University Charsadda, Khyber Pakhtunkhwa, Pakistan, 4 Department of Biotechnology, University of Swabi, Anbar, Khyber Pakhtunkhwa, Pakistan, 5 Department of Bioinformatics, Shaheed Benazir Bhutto Women University Peshawar, Khyber Pakhtunkhwa, Pakistan, 6 Department of Microbiology, The University of Haripur, Khyber Pakhtunkhwa, Pakistan, 7 Department of Agronomy, Faculty of Agriculture and Environment, Islamia University Bahawalpur, Bahawalpur, Pakistan

* m.israr@uoh.edu.pk

## Abstract

### Background & aim

Hepatitis B and C infections are global issues that are associated with a massive financial burden in developing countries where vertical transmission is the major mode and remains high. This cross-sectional study was designed to investigate the seroepidemiology and associated risk factors of hepatitis B virus (HBV) and hepatitis C virus (HCV) infections among 375 pregnant women attending antenatal care health facilities at Bacha Khan Medical Complex (BKMC) Shahmansoor and District Head Quarter (DHQ) Hospital Swabi, Khyber Pakhtunkhwa, Pakistan.

### Methodology

From a total of 375 pregnant women selected using systematic random sampling from both hospitals, 10 ml of blood samples were collected and alienated serum was examined for indicators identification through the Immuno-Chromatographic Test (ICT) and 3rd Generation Enzyme-Linked Immunosorbent Assay (ELISA). A pre-structured questionnaire was used to collect the socio-demographic data and possible risk factors. The data was analyzed via SPSS 23.0 statistical software. A chi-square analysis was performed to determine the association between variables. P-value < 0.05 was set statistically significant.

### Results

The overall frequency of HBV and HCV among 375 pregnant women involved in the study was 3.7% and 2.1% respectively. None of the pregnant women were co-infected with HBV

**Data Availability Statement:** All relevant data are within the manuscript and its Supporting Information files.

**Funding:** The authors received no specific funding for this work.

**Competing interests:** The authors have declared that no competing interests exist.

and HCV. Dental extraction (P = 0.001) and blood transfusion (P = 0.0005) were significantly allied with HBV infection while surgical procedure (P = 0.0001) was significantly associated with HCV infection. Moreover the sociodemographic characteristics: residential status (P = 0.017) and educational level (P = 0.048) were found significant risk factors of HBsAg and maternal age (P = 0.033) of anti-HCV, respectively.

## Conclusion & recommendation

HBV and HCV infections are intermediary endemic in the study area. A higher prevalence of HBV was detected among pregnant mothers with a history of dental extraction, history of blood transfusion, resident to the urban area and low educational level. The age and surgical procedures were the potential risk factors found significantly associated with HCV positivity among pregnant mothers in our setup. Future negotiations to control vertical transmission should include routine antenatal screening for these infections early in pregnancy and the requirement of efficient preventive tools including the birth dose of the hepatitis B vaccine in combination with hepatitis B immune globulins to the neonate.

## Introduction

Hepatitis B and C viruses are members of *Hepadnaviridae* and *Flaviviridae* families of the virus respectively that cause liver infections in humans [1]. These viral infections are characteristically linked with cirrhosis, chronic hepatitis, and hepatocellular carcinoma that are the leading cause of morbidity and mortality in the population of developing countries including Pakistan [2]. As per 2017 WHO report, about 257 million individuals are living with hepatitis B virus infection and approximately 3% of the world's people are infected with chronic HCV in which the highest frequency rate is reported from Africa [3].

In Pakistan, the scenario is worse than the developed nations and up till now HBV and HCV infected approximately 12 million people, which shows 7.4% prevalence, of which 2.4% are infected with HBV and 4.9% are infected with HCV [4]. The prevalence of these viral infections in pregnant women is also at high risk which is about 3.16% and 4.65%, respectively of the country population [5, 6] and increasing rapidly.

Previously, it has been anticipated that during pregnancy viral hepatitis is often associated with the development of hepatocellular carcinoma through estrogen secretion which leads to increased maternal mortality. Moreover, during pregnancy, the comprehensive immune containment also contributes to the enlargement of malignancy [7]. The transmission routes for both HBV and HCV viruses are of great apprehension especially the vertical transmission from mother to offspring [8]. Among them, HBV is transmitting through mucosal or parenteral contact to body fluids and infected blood, normally either by a horizontal or vertical transmission route during infancy in exceedingly endemic areas which results in an elevated rate of chronic infections [8]. During delivery, approximately 90% of the children infected with HBV have a high risk of becoming a chronic carrier and about 15 to 25% chances of growing hepatocellular carcinoma during old age which ultimately leads to mortality [8]. The prenatal transmission rate of both the HBV and HCV viruses is about 10 and 5%, respectively. The maternal transmission of HBV infection can be reduced from 85 to 95% through a dose of Hepatitis B vaccine in combination with hepatitis B immune globulins to the neonate [8].

The epidemiology of viral hepatitis infections in a population can be anticipated by the risk factors such as ear/nose piercing, tattooing on the body, dental extraction, surgical procedure, history of abortion, history of sexually transmitted disease (STD), shaving eyebrow, body piercing for treatment, delivery by TBA, receiving a blood transfusion, multiple sexual partners, history of contact with a jaundiced patient, injections and vertical transmission [9].

Nowadays, routine antenatal HBV and HCV screening of mothers during pregnancy become an important area of public health concern worldwide to prevent vertical transmission of these viral infections, which are the fundamental cause of maternal death. With this in mind, the present cross-sectional study was designed to investigate the seroepidemiology and the possible associated risk factors of HBV and HCV among pregnant women attending antenatal care services at selected Hospitals: Bacha Khan Medical Complex (BKMC) Shahmansoor and DHQ Hospital in Swabi which is a less developed district of Khyber Pakhtunkhwa, Pakistan.

## Materials and methods

### Study design and setting

This seroepidemiologic cross-sectional study was conducted at the Bacha Khan Medical Complex (BKMC) Shahmansoor and District Head Quarter (DHQ) Hospital Swabi which are facilitated with 40 and 20 beds of Antenatal Care (ANC) respectively, for Gynecology and Obstetrics. Both the BKMC and DHQ Hospitals are declared Teaching Hospitals of Gajju Khan Medical College (GKMC) Swabi and are receiving approximately 60–90 and 30–60 pregnant women per day respectively, from the surrounding urban and rural areas of District Swabi, Khyber Pakhtunkhwa Pakistan.

### Source population

All pregnant women attending hospital maternity wards for antenatal care at Bacha Khan Medical Complex (BKMC) Shahmansoor and District Head Quarter (DHQ) Hospital Swabi from surrounding urban and rural areas of district Swabi were the source population.

### Study population

All pregnant women attending hospital maternity wards for antenatal care at Bacha Khan Medical Complex (BKMC) Shahmansoor and District Head Quarter (DHQ) Hospital Swabi from surrounding urban and rural areas of district Swabi during July 2019 -January 2020 were the study population.

### Sample size and sampling technique

A sample size of 375 pregnant women (200 from BKMC Shahmansoor and 175 from DHQ Hospital Swabi) was calculated based on 95% confidence level, 0.05 margins of error, and 10.5% of HBV and HCV seroprevalence. All the participants were recruited using systematic random sampling technique.

### Inclusion and exclusion criteria

The pregnant mothers whose pregnancy was confirmed by an ultrasound scan were included in study. Pregnant mothers who were critically sick and unable to answer the questionnaire during data collection were excluded from study.

## Study variables

**Dependent variables.** Seroprevalence of HBV and HCV.

**Independent variables.** History of sexually transmitted disease (STD), tattooing on body, dental extraction, abortion, shaving eyebrow, ex-delivery at health facility, surgical procedure, hospital admission, receiving blood transfusion, history of visiting abroad, maternal age, residential status, educational level, family monthly income, occupation and parity.

## Data collection

In this study, we included data on associated risk factors and socio-demographic characteristics of the participated pregnant women, and data about HBV and HCV seroprevalence from blood samples. Trained antenatal care health nurses were assigned to collect the data on socio-demographic and risk factors through face to face interview using a pre-structured questionnaire.

## Sample collection

Ten milliliter of blood samples were collected from each participated mother under the aseptic condition and was kept at room temperature for 30 minutes to facilitate clotting. The clotted blood was then centrifuged at 4000 rpm for five minutes to separate the serum. Each sample was alienated into two aliquot parts; one was used for HBsAg detection and the other was used for anti-HCV antibody screening per company instructions. The serum samples were stored in the refrigerator at -20°C until transferred to the pathology laboratory for serologic screening.

## Serologic screening

For initial qualitative detection of HBsAg and HCV Ab, ICT strips (Acon USA) were used. The sensitivity and specificity of both the strips are above 99% and 98%, respectively. All the positive samples on ICT were further confirmed by 3rd Generation Enzyme- Linked Immunosorbent Assay (ELISA) (EASE BN-96 TMB, Taiwan) as previously described [10, 11].

## HBsAg and HCV-Ab detection through ICT

HBsAg and Anti-HCV antibodies were detected through ICT strips (Acon USA) following the company instructions. The strip was detached from the foil pouch and was placed on a hygienic, dried surface. Then 5 μL of serum each for HBsAg and HCV-Ab detection was decanted in the strip and was dispensed with two drops of a buffer. After 15 min, the results were interpreted according to the appearance of color bands. To check the validity of the test strip, a control group was also run. In both test and control bands, a purplish-red color appeared on the membrane of the strip which confirmed a positive result. One red line appears in the layer of the strip in the control region (C). The appearance of no red line in the test area indicated a negative result [10].

## HBsAg and HCV-Ab detection through ELISA

HBsAg and Anti-HCV antibodies were detected through 3rd generation ELISA (EASE BN-96 TMB, Taiwan) as per company instructions. Three wells pre-coated with HBsAg and anti-HCV antigens each were taken and kept in a holder. 50 μL of specimens, positive control and negative control were dispensed in their specific wells. Then 50 μL of horse-reddish peroxidase conjugate (HRP- conjugate) was added to each well except the blank and was mixed by pattering the plate smoothly. Enclosed the plate with glue slip and was incubated at 37°C for one hour. After incubation, the glue slip was detached from each well and washed five times with a

diluted buffer. 50 μL of chromogenic solution A and 50 μL of chromogenic solution B were dispensed into each well including the blank and were mixed by pattering the plate smoothly for 15 seconds. The plate was then incubated at 37˚C in the dark for 15 min without shaking. 50 μL of stop-solution was added to stop the reaction. The absorbance of specimens and controls was determined within 15 min by spectrophotometer at 492 nm. The enzymatic reaction between the HRP-conjugate and chromogenic solutions forms a blue color in HBsAg and HCV-Ab positive sample wells and positive control well before the addition of the stop solution. After adding the stop solution, the blue color in HBsAg and HCV-Ab positive wells and positive control well altered to yellow color; Negative samples have a clear water-like appearance before and after the dispensing of the stop solution. The sample with absorbance value greater than or equal to the cut-off value i.e. (2.00) was considered reactive for HBsAg and HCV-Ab while the sample with absorbance value less than the cut-off value was considered HBsAg and HCV-Ab negative [11, 12].

## Data quality assurance

The questionnaire was pre-tested on a 10% sample, and validated by Cronbach's alpha test before the actual data collection. Nurses were trained for one day with practice before data collection. Regular supportive supervision was given to nurses for data collection.

The WHO and national guidelines were followed to collect, process, and test serum samples. Data completeness and consistency were checked daily, during data entry and analysis.

## Statistical analysis

The obtained questionnaire from participated pregnant mother was implied into Statistical Package for Social Sciences (SPSS) software, version 23.0 by the principal investigator. The data were cross-checked for wrong and omitted entries by computing descriptive statistics for all the variables. Normality of the data was checked using Shapiro-Wilk test. Chi square test was used for association between HBV and HCV sero status with characteristics of the pregnant mothers. The p-values less than 0.05 were set statistically significant.

## Ethical endorsement

The ethical endorsement for study conduction was first approved by the Institutional Research Ethical Committee (IREC) of the Department of Biology, The University of Haripur (MS-E-tics-099/2019-2020) and finally official permission was obtained from the administration of BKMC and DHQ Hospital Swabi to carry out the study in the departments of Obstetrics and Gynecology. The study was conducted according to declaration of Helsinki. Both verbal and written informed consent was obtained from all pregnant mothers. They were informed that participation was voluntary and they were at authorization to withdraw from the study at any time without any consequences to them. They were told 10 ml of their blood would be drawn for the HBsAg and Anti-HCV antibodies screening. The test results of HBsAg and HCV-Ab positive participants was communicated with respective physician for further treatment. All the results were kept confidential and remaining blood samples were discarded and did not use for any other purpose.

## Results

### Socio-demographic characteristics

A total of 375 pregnant women participated in the study making the response rate of 100%. The majority of the participated women 211 (56.2%) were in the rage of 20–29 years followed

by 135 (36.0%) in the range of 30–39 years and 29 (7.7%) were in the age of 40 years or above 40 years. Two hundred and thirty (61.6%) of the pregnant mothers were rural dwellers. Regarding level of education, 167 (44.5%) of the women learned to the level of secondary school and above, 111 (29.6%) learned to primary level whereas 97 (25.8%) had no formal education. The majority of the pregnant mothers were housewives that account 209 (55.7%) followed by employees 166 (44.2%). Concerning family monthly income, 177 (47.2%) of the participants had monthly family income of Rs 15000 and below followed by 143 (38.1%) with monthly income of Rs 15000–20000 and 55 (14.6%) had a family monthly income of Rs 20000 and above. Regarding the number of parity, 274 (73.0%) of the mothers had primigravida (first time birth) followed by 68 (18.1%) had a second pregnancy and 33 (8.8%) had multigravida (more than one time pregnancy) as shown in Table 2.

## Associated risk factors of HBV and HCV

In this study, a total of 10 associated risk factors with HBV and HCV such as history of STD, tattooing on body, dental extraction, abortion, shaving eyebrow, ex-delivery at health facility, surgical procedure, hospital admission, receiving blood transfusion and history of visiting abroad were studied. From a total of 375 study participants, 2 (3.5%) of HBV and 2 (3.5%) of HCV positive women had a history of sexually transmitted disease (STD), 3 (5.2%) of HBV and 1 (1.7%) of HCV positive women had tattooing on their bodies, 4 (19.1%) of HBV and 1 (3.8%) of HCV positive women had history of dental extraction, 1 (5.5%) of HBV and 1 (5.5%) of HCV positive women had history of abortion, 2 (5.7%) of HBV and 0 (0.00%) of HCV positive women had routine for shaving their eyebrow, 4 (7.0%) of HBV and 3 (5.2%) of HCV positive women had ex-delivery at health facility, 1 (2.7%) of HBV and 4 (10.8%) of HCV positive women had history of surgical procedure, 4 (7.1%) of HBV and 2 (3.5%) of HCV positive women had history of hospital admission, 5 (14.2%) of HBV and 1 (2.8%) of HCV positive women had history of receiving blood transfusion, while the women with history of visiting abroad had no association with HBV and HCV seropositivity as shown in Table 1.

## Prevalence of HBV and HCV infections

The overall prevalence of HBV and HCV detected through ICT and ELISA was 3.7% and 2.1% respectively as shown in Fig 1.

However, the rest of the factors were found to have no significant relation with HBV prevalence. The surgical procedure (P = 0.0001) (Table 1) was found a significantly associated risk factor with HCV sero-positivity. Similarly, the sero-positivity of HCV was confined to mothers of the age group from 20–29 years. Statistically, there exists a significant disparity in the association between seroprevalence of HCV and maternal age (P = 0.033) (Table 2) however, rest of the risk factors showed no significance with HCV seropositivity.

## Discussion

Hepatitis B and C viral infections are global issues that are associated with a massive financial burden both in developing and developed countries [13]. In Pakistan, the situation is worse than in the rest of the world where treatments of these viral infections in pregnant mothers are contraindicated during pregnancy due to the prospective risks of the diagnostic procedure that are the major source of morbidity and mortality among them [2] and which needs serious consideration to implement routine antenatal screening of pregnant women to be prevented from vertical transmission.

In this study, an overall prevalence of HBV among pregnant mothers was 3.7% which shows intermediate endemicity of HBV infection according to world health organization

**Table 1. HBV and HCV associated risk factors among pregnant women attending maternity wards at selected Hospitals, DHQ Hospital Swabi and BKMC Shahmansoor.**

| Risk factors | Response | HBV Sero-status | | $X^2$ | P-Value | HCV Sero-status | | $X^2$ | P-Value |
|---|---|---|---|---|---|---|---|---|---|
| | | +Ve n (%) | – Ve n (%) | | | +Ve n (%) | – Ve n (%) | | |
| History of STD | Yes | 2(3.5) | 55 (96.5) | 0.01 | 0.922 | 2 (3.5) | 55(96.5) | 0.61 | 0.435 |
| | No | 12(3.7) | 306 (96.2) | | | 6 (1.8) | 312(98.1) | | |
| Tattooing on body | Yes | 3(5.2) | 55 (94.8) | 3.08 | 0.079 | 1 (1.7) | 57(98.2) | 0.06 | 0.814 |
| | No | 11(3.4) | 307(96.5) | | | 7(2.2) | 310(97.7) | | |
| Dental extraction | Yes | 4(19.1) | 22(84.6) | 10.55 | 0.001 * | 1(3.8) | 25(96.11) | 0.39 | 0.531 |
| | No | 10(2.8) | 339(97.1) | | | 7(2.0) | 342(97.9) | | |
| Abortion | Yes | 1(5.5) | 17(94.4) | 0.17 | 0.676 | 1(5.5) | 17(94.4) | 1.06 | 0.303 |
| | No | 13(3.6) | 344(96.3) | | | 7(1.9) | 350(98.0) | | |
| Shaving eyebrow | Yes | 2(5.7) | 50(96. 1) | 0.03 | 0.961 | 0(0.00) | 52(100) | 1.32 | 0.251 |
| | No | 12(3.7) | 311(96.2) | | | 8(2.5) | 315(97.5) | | |
| Ex-delivery at health facility | Yes | 4(7.0) | 53(92.9) | 2.02 | 0.155 | 3(5.2) | 54(94.7) | 3.15 | 3.15 |
| | No | 10(3.1) | 308(96.8) | | | 5(1.5) | 313(98.4) | | |
| Surgical procedure | Yes | 1(2.7) | 36(97.3) | 0.12 | 0.729 | 4(10.8) | 33(89.1) | 14.81 | 0.0001 * |
| | No | 13(3.8) | 326(96.1) | | | 4(1.1) | 334(98.8) | | |
| Hospital admission | Yes | 4(7.1) | 52(92.8) | 2.13 | 0.144 | 2(3.5) | 54(96.4) | 0.65 | 0.419 |
| | No | 10(3.1) | 309(96.8) | | | 6(1.8) | 313(98.1) | | |
| Receiving blood transfusion | Yes | 5(14.2) | 30(85.7) | 11.96 | 0.0005 * | 1(2.8) | 34(97.1) | 0.10 | 0.755 |
| | No | 9(2.6) | 331(97.3) | | | 7(2.0) | 333(97.9) | | |
| History of visiting abroad | Yes | 0(0.00) | 8(100) | 0.32 | 0.573 | 0(0.00) | 8(100) | 0.18 | 0.672 |
| | No | 14(3.8) | 353(96.1) | | | 8(2.1) | 359(97.8) | | |

STD: Sexually transmitted disease.

classification criteria [14].This prevalence is comparable and in agreement with a previous study reported from Swat in which the prevalence of HBV among pregnant mothers was 3.98% [15]. This 3.7% prevalence of HBV in our study was higher than that found in the previous study among pregnant mothers of 0.34% [16]. However, an earlier reported prevalence for Hepatitis B of 12.3% in pregnant women was very high compared to our study [17]. The similar high prevalence of HBV among pregnant mothers compared to our study was also reported by Ugbebor *et al.*, 2011[18]. This variation in results between the studies might be due to an unhygienic environment, low socioeconomic conditions, lack of awareness, and divergence in the geographical distribution among the countries.

Hepatitis C among pregnant women is also on the raise during pregnancy which is an alarming problem in Pakistan that needs to be considered. The various studies conducted in Pakistan described the prevalence of HCV ranging from 0.7% to 20% [19]. In our study, 2.1% of the pregnant mothers were found positive for HCV which shows intermediate endemicity according to world health organization classification criteria [15]. This 2.1% prevalence of HCV is quite lower than the previously reported prevalence of 11.68% [20]. This difference might be due to lack of knowledge among women, the different diagnostic procedures utilized for the screening of viral infection and poor socioeconomic conditions among the people of this country. However, 2.2% of HCV prevalence previously found among pregnant women of district Haripur is an agreement with the findings of the current study [21]. The prevalence of HCV in our study was higher when compared with the earlier reported prevalence of 0.69% [16]. A similar finding reported by a study in India was 1.03% which is slightly low as compared to the current study [22]. Other comparable

# Sero-status of HBsAg and HCV-Ab

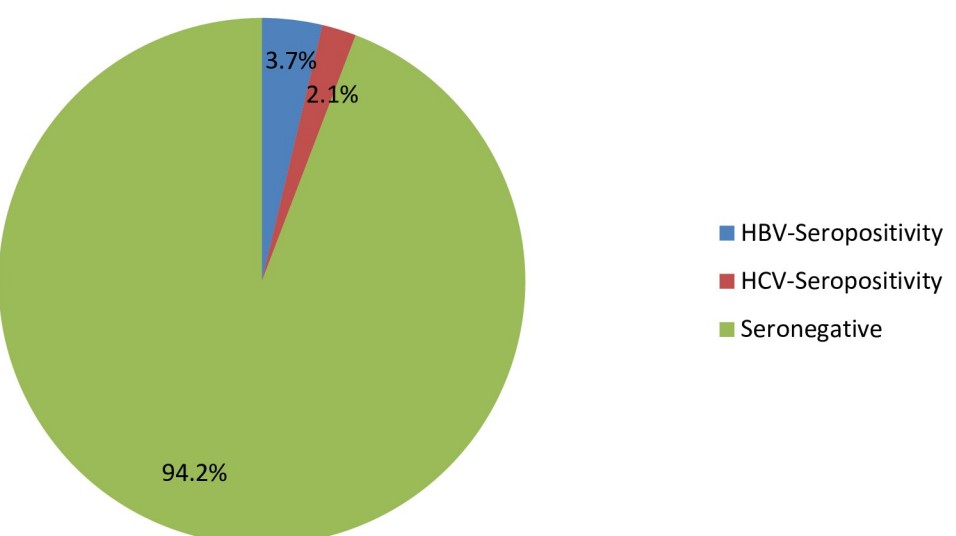

**Fig 1. Seroprevalence of HBsAg and HCV-Ab among pregnant women attending maternity wards at BKMC Shahmansoor and DHQ Hospital Swabi, Khyber Pakhtunkhwa, Pakistan.** None of the mothers were co-infected with HBV and HCV. The chi-square analysis of the association between different risk factors and the prevalence of HBV and HCV revealed that dental extraction (P = 0.001) and blood transfusion (P = 0.0005) (Table 1) were the significant risk factors associated with HBV prevalence. Correspondingly, the sociodemographic characteristics of pregnant mothers such as residential status (P = 0.017) and educational level (P = 0.048) (Table 2) were also significantly allied with HBV infection.

**Table 2. Prevalence of HBV and HCV infection in association to sociodemographic characteristics of pregnant women attending maternity wards at selected Hospitals, DHQ Hospital Swabi and BKMC Shahmansoor.**

| Characteristics | | Total n (%) | HBV Infection | | HCV Infection | |
|---|---|---|---|---|---|---|
| | | | +Ve n (%) | P-Value | +Ve n (%) | P-Value |
| Maternal age (years) | 20–29 | 211(56.2) | 7(3.3) | 0.853 | 5(2.3) | 0.033 * |
| | 30–39 | 135(36.0) | 5(3.7) | 0.470 | 2(1 14) | |
| | ≥40 | 29(7.7) | 2(6.8) | | 1(3.4) | |
| Residential status | Urban | 145(38.6) | 4(2.7) | 0.017 * | 2(1.4) | 0.743 |
| | Rural | 230(61.6) | 10(4.3) | | 6(2.6) | |
| Educational level | Illiterate | 97(25.8) | 5(5.2) | 0.048 * | 3(3.1) | 0.494 |
| | Primary | 111(29.6) | 6(5.4) | 0.124 | 4(3.6) | 0.536 |
| | Secondary and above | 167(44.5) | 3(1.8) | | 1(0.5) | |
| Family monthly income (Rs) | ≤15000 | 177(47.2) | 7(3.9) | 0.836 | 5(2.8) | 0.643 |
| | 15000–20000 | 143(38.1) | 5(3.5) | 0.963 | 2(1.4) | 0.727 |
| | ≥20000 | 55(14.6) | 2(3.6) | | 1(1.8) | |
| Occupation | Employed | 166(44.2) | 5(3.0) | 0.528 | 3(1.8) | 0.924 |
| | Housewives | 209(55.7) | 9(4.3) | | 5(2.4) | |
| Parity | Primigravida | 274(73.0) | 9(3.3) | 0.344 | 6(2.2) | 0.732 |
| | Gravidity | 68(18.1) | 4(5.8) | 0.560 | 2(2.9) | 0.626 |
| | Multigravida | 33(8.8) | 1(3.0) | | 0(0.0) | |

seroepidemiologic studies were also reported with a low prevalence of HCV in their particular setup at different times [23, 24].

The current study demonstrated that the pregnant mothers with a history of blood transfusion are significantly associated with HBsAg seropositivity, which is following the findings reported by Abongwa et al. (2016) [25]. Similarly, studies reported from African countries such as Nigeria, Cameroon, and Sudan also observed that the frequency of HBV perceived among mothers with a history of blood transfusion had a significant statistical association [25–27]. This resemblance in findings is because blood transfusion is a well- recognized risk factor for HBsAg, and the occurrence of viral infection after one pint of blood is almost analogous to after several blood transfusions [28]. Similarly, a history of dental extraction was found to be a possible risk factor for HBV infection in our study.

Consequently, having a history of tooth extraction among pregnant women augmented the possibility of HBV occurrence compared to their counterparts. Comparing these results to other studies, a consistent finding was carried out by Awole and Gebre-Selassie (2005) [29] and Mollaa et al. (2015) [30].This might be due to non- observance of the procedure on infection control and the utilization of reusable or non-disposable equipment and the lack of adequate sterilization technology. In our study, the pregnant mothers with sociodemographic characteristics i.e. low educational level and residency to the urban area were also significantly associated with HBV seropositivity which is comparable to the study conducted in Ethiopia [31].

The current study investigated that pregnant mothers with a history of surgical procedures are significantly associated with HCV infection which is consistent with the findings of Muzaffar et al. (2009) [32] and Akhtar et al. (2014) [20]. Moreover, a comparable supporting study was also reported by Jaffery et al. (2005) conducted at Shifa International Hospital Islamabad [33]. Likewise, the age aof pregnant mothers is also a well-known risk factor for HCV infection which frequency is rising to the age of 40 years and then declines over time [34]. The highest prevalence of HCV mostly occurs among women of reproductive ages [35]. This high seroprevalence might be due to the high probability of exposure of these pregnant women to risk factors and also their high fertility rate during which they give birth to more children. In the present study, the high HCV prevalence was observed from the age group of 20–29 years (P-value 0.03, Table 2), which is per study reported by Ishaq et al.(2011) in Kuwait Teaching Hospital Peshawar (21–29 years) [36], Jilani et al. (2017) in Karachi (26–30 years) [37] and Gul et al. (2009) in Ayub medical college (25–35 years) [38]. Almost similar age groups associated with high HCV prevalence were also reported by Kumar et al. (2007) in India [22] and Prasad et al. (2007) in Switzerland [39].

## Study limitations

This study faces some limitations. First, the sample size is too small for generalization. Secondly, the small number of seropositivity of HBV and HCV made it difficult to establish associations. Despite, the screening method used in this study (i.e. ICT and ELISA) have a relative sensitivity and specificity of above 99% when compared to the confirmatory test and so can give accurate results. Moreover, the same screening methods were adopted previously [10–12].

## Conclusion

HBV and HCV infections are intermediary endemic in the study area. A higher prevalence of HBV was detected among pregnant mothers with a history of dental extraction, history of blood transfusion, resident to the urban area, and low educational level. The age and surgical

procedures were the potential risk factors found significantly associated with HCV positivity among pregnant mothers in our setup.

## Recommendation

Future negotiations to control vertical transmission should include routine antenatal screening for these infections early in pregnancy and the requirement of efficient preventive tools including the birth dose of the hepatitis B vaccine in combination with hepatitis B Immunoglobulins to the neonate.

## Supporting information

**S1 File. Questionnaire.**
(DOCX)

## Acknowledgments

We are very grateful to the staff members of BKMC Shahmansoor and DHQ Hospital Swabi for their constructive suggestions, technical support and guidance during the study duration. We are also grateful to the participants in our study setup.

## Author Contributions

**Conceptualization:** Muhammad Israr.

**Data curation:** Muhammad Israr, Muhammad Idrees, Bashir Ahmad, Rashid Iqbal.

**Formal analysis:** Arif Nawaz, Muhammad Idrees, Aishma Khattak, Shafiq Ur Rehman, Azizullah Azizullah, Bashir Ahmad, Rashid Iqbal.

**Funding acquisition:** Arif Nawaz.

**Investigation:** Fawad Ali, Arif Nawaz.

**Software:** Fawad Ali.

**Supervision:** Muhammad Israr.

**Validation:** Rashid Iqbal.

**Writing – original draft:** Muhammad Israr.

**Writing – review & editing:** Muhammad Israr, Shafiq Ur Rehman, Bashir Ahmad, Syeda Asma Bano.

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
