## [Decision Letter · Decision Letter 0]

29 Apr 2021

PONE-D-21-11040

Seroepidemiology and Associated Risk Factors of Hepatitis B and C Virus Infections
among Pregnant Women Attending Maternity Wards at Two Hospitals in Swabi, Khyber
Pakhtunkhwa, Pakistan

PLOS ONE

Dear Dr. Israr,

Thank you for submitting your manuscript to PLOS ONE. After careful consideration, we
feel that it has merit but does not fully meet PLOS ONE’s publication criteria as it
currently stands. Therefore, we invite you to submit a revised version of the
manuscript that addresses the points raised during the review process. More details
are required in the results sections (see review) , precisions in the Material and
methods, discussion may be strenghtened.

If you would like to make changes to your financial disclosure, please include your
updated statement in your cover letter. Guidelines for resubmitting your figure
files are available below the reviewer comments at the end of this letter.

We look forward to receiving your revised manuscript.

Kind regards,

Isabelle Chemin, PhD

Academic Editor

PLOS ONE

Journal Requirements:

PLOS requires an ORCID iD for the corresponding author in Editorial
Manager on papers submitted after December 6th, 2016. Please ensure that
you have an ORCID iD and that it is validated in Editorial Manager. To
do this, go to ‘Update my Information’ (in the upper left-hand corner of
the main menu), and click on the Fetch/Validate link next to the ORCID
field. This will take you to the ORCID site and allow you to create a
new iD or authenticate a pre-existing iD in Editorial Manager. Please
see the following video for instructions on linking an ORCID iD to your
Editorial Manager account: https://www.youtube.com/watch?v=_xcclfuvtxQ

3. Thank you for including your ethics statement: "The ethical endorsement for study
conduction was approved by the Institutional Research Ethical Committee (IREC) of
BKMC and DHQ Hospital Swabi."

a) Please provide additional details regarding participant consent. In the ethics
statement in the Methods and online submission information, please ensure that you
have specified (1) whether consent was informed and (2) what type you obtained (for
instance, written or verbal, and if verbal, how it was documented and witnessed). If
your study included minors, state whether you obtained consent from parents or
guardians. If the need for consent was waived by the ethics committee, please
include this information.

4. We suggest you thoroughly copyedit your manuscript for language usage, spelling,
and grammar. If you do not know anyone who can help you do this, you may wish to
consider employing a professional scientific editing service. 

6. Please include your tables as part of your main manuscript and remove the
individual files. Please note that supplementary tables (should remain/ be uploaded)
as separate "supporting information" files

Reviewers' comments:

Reviewer's Responses to Questions

**Comments to the Author**

1. Is the manuscript technically sound, and do the data support the conclusions?

Reviewer #1: Yes

2. Has the statistical analysis been performed
appropriately and rigorously? 

Reviewer #1: Yes

3. Have the authors made all data underlying the
findings in their manuscript fully available?

Reviewer #1: Yes

4. Is the manuscript presented in an intelligible
fashion and written in standard English?

Reviewer #1: Yes

5. Review Comments to the Author

Reviewer #1: Title: Seroepidemiology and Associated Risk Factors of Hepatitis B and C
Virus Infections among Pregnant Women Attending Maternity Wards at Two Hospitals in
Swabi, Khyber Pakhtunkhwa, Pakistan

In the abstract: there is no result conclusion indicated!

Method part

Source population is not stated under method part

Under method part: hierarchal permission process for the research was not
described well and ethical principle was not stated well! (e.g. for those positive
cases how the investigator manage the issue and also whether verbal or written
consent was taken)

Sampling techniques and sample size were not described.

Quality assurance measures were not stated.

Hospital and health center were used interchangeably (but not the
same)

Interpretation of the laboratory test for both viruses was not
stated!

Data analysis steps were not clear to understand how it was performed

Result

Result part was not elaborated well (e.g by describing the
socio-demographic factors except in the table)

Prevalence of HBV and HCV, the socio-demographic factors and associated
factors shall be stated separately under different subtitles as the objective

In addition to table, if graphs are used to show the result, it will be
good to easily understand the result in %.

There must be an explanation for why “Dental extraction, Receiving blood
transfusion, Educational level and Residential status associated with HBVsAg
positivity and Surgical procedure, maternal age for HCV Ab positivity”

Under discussion:

Strength and Limitation of the study shall be specified

Under Conclusion

The result was not interpreted well whether it was high or low according
to WHO guidelines for HBV and HCV (whether it falls in the endemic interval or hyper
endemic interval and only risk factors were stated in line with the result.

6. PLOS authors have the option to publish the peer
review history of their article (what does this mean?). If published, this will
include your full peer review and any attached files.

If you choose “no”, your identity will remain anonymous but your review may still be
made public.

**Do you want your identity to be public for this peer review?** For
information about this choice, including consent withdrawal, please see our
Privacy Policy.

Reviewer #1: No

comment April 18 2021.docx
---

## [Author Response · Author response to Decision Letter 0]

21 May 2021

Dear Dr. Isabelle Chemin, 

We are grateful to you and the reviewers for reading and constructive suggestions
which improved our manuscript and for the opportunity to revise and resubmit. We are
pleased to submit the revised manuscript entitled “Seroepidemiology and Associated
Risk Factors of Hepatitis B and C Virus Infections among Pregnant Women Attending
Maternity Wards at Two Hospitals in Swabi, Khyber Pakhtunkhwa, Pakistan” with Id
PONE-D-21-11040 for consideration in PLOS ONE. On the following pages, you will find
our response to the editor and reviewer comments. The editor and reviewer comments
are highlighted with red color and responses to each point with bold text. On behalf
of my co-authors, I thank you for your consideration of this resubmission. We
appreciate your time and look forward to your response. 

Sincerely,

Dr. Muhammad Israr, PhD, Postdoc (Corresponding Author)

Associate Professor, Department of Biology 

The University of Haripur, KPK, Pakistan 

Responses to Editor Comments:

Thank you for submitting your manuscript to PLOS ONE. After careful consideration, we
feel that it has merit but does not fully meet PLOS ONE’s publication criteria as it
currently stands. Therefore, we invite you to submit a revised version of the
manuscript that addresses the points raised during the review process. More details
are required in the results sections (see review); precisions in the Material and
methods, discussion may be strengthened.

We are grateful to the editor for these constructive suggestions. The results,
material and method and discussion sections were revised according to editor
suggestions and highlighted with red color.

• A rebuttal letter that responds to each point raised by the academic editor and
reviewer(s). You should upload this letter as a separate file labeled 'Response to
Reviewers'

• A marked-up copy of your manuscript that highlights changes made to the original
version. You should upload this as a separate file labeled 'Revised Manuscript with
Track Changes'

A letter “Response to Reviewers” Revised Manuscript with Track Changes and Revised
Manuscript without Track Changes are attached according to editor suggestions.

The whole manuscript is revised according to the journal PLOS ONE style
requirements

2. PLOS requires an ORCID iD for the corresponding author in Editorial Manager on
papers submitted after December 6th, 2016. Please ensure that you have an ORCID iD
and that it is validated in Editorial Manager. To do this, go to ‘Update my
Information’ (in the upper left-hand corner of the main menu), and click on the
Fetch/Validate link next to the ORCID field. This will take you to the ORCID site
and allow you to create a new iD or authenticate a pre-existing iD in Editorial
Manager. Please see the following video for instructions on linking an ORCID iD to
your Editorial Manager account:

https://www.youtube.com/watch?v=_xcclfuvtxQ

 The registered ORCID ID of the corresponding author is validated in the Editorial
Manager.

3. Thank you for including your ethics statement: "The ethical endorsement for study
conduction was approved by the Institutional Research Ethical Committee (IREC) of
BKMC and DHQ Hospital Swabi." 

a) Please provide additional details regarding participant consent. In the ethics
statement in the Methods and online submission information, please ensure that you
have specified (1) whether consent was informed and (2) what type you obtained (for
instance, written or verbal, and if verbal, how it was documented and witnessed). If
your study included minors, state whether you obtained consent from parents or
guardians. If the need for consent was waived by the ethics committee, please
include this information.

For additional information about PLOS ONE ethical requirements for human subject’s
research, please refer to http://journals.plos.org/plosone/s/submission-guidelines#loc-human-subjects-research.

The ethical approval was modified according to the editor suggestions both in Method
section and in the online manuscript submission system. 

4. We suggest you thoroughly copyedit your manuscript for language usage, spelling,
and grammar. If you do not know anyone who can help you do this, you may wish to
consider employing a professional scientific editing service. 

Upon resubmission, please provide the following

Thank you for this comment. The manuscript is copyedited for language usage,
spelling, and grammar by an English speaking colleague Dr. Helen Appleton, Professor
at the University of Oxford. 

a) Please clarify the sources of funding (financial or material support) for your
study. List the grants or organizations that supported your study, including funding
received from your institution

b) State what role the funders took in the study. If the funders had no role in your
study, please state: “The funders had no role in study design, data collection and
analysis, decision to publish, or preparation of the manuscript.

c) If any authors received a salary from any of your funders, please state which
authors and which funders

The amended statements for the funders and funding source in the research work are
given bellow 

b) “The funders had no role in study design, data collection and analysis, decision
to publish, or preparation of the manuscript”.

d) “The authors received no specific funding for this work.”

6. Please include your tables as part of your main manuscript and remove the
individual files. Please note that supplementary tables (should remain/ be uploaded)
as separate "supporting information" files

The authors are grateful to the editor for this comment. The tables and figures were
included in the main manuscript and the individual files were removed according to
editor suggestion.

The references list was modified and rearranged in the revised manuscript according
to journal references format as suggested by the editor. 

Responses to Reviewer comments 

Comments to the Author

1. Is the manuscript technically sound, and do the data support the conclusions?

Reviewer #1: Yes________________________________________2. Has the statistical
analysis been performed appropriately and rigorously?

Reviewer #1: Yes________________________________________3. Have the authors made all
data underlying the findings in their manuscript fully available?

Reviewer #1: Yes________________________________________4. Is the manuscript
presented in an intelligible fashion and written in standard English?

Reviewer #1: Yes

5. Review Comments to the Author

Reviewer #1: Title: Seroepidemiology and Associated Risk Factors of Hepatitis B and C
Virus Infections among Pregnant Women Attending Maternity Wards at Two Hospitals in
Swabi, Khyber Pakhtunkhwa, Pakistan

In the abstract: there is no result conclusion indicated!

We are grateful to the reviewer for this comment. The abstract of the manuscript is
divided into subsections i.e. (Background & aim, Methodology, Results,
Conclusion and Recommendation) and modified according to journal format. The result
conclusion is also indicated according to reviewer comment and highlighted with red
color.

Method part

Source population is not stated under method part

The authors are thankful to reviewer for this comment. Source population is added in
the method section under the subheading “Source and study population” and
highlighted with red color.

Under method part: hierarchal permission process for the research was not
described well and ethical principle was not stated well! (e.g. for those positive
cases how the investigator manage the issue and also whether verbal or written
consent was taken).

Thanks for this comments. The hierarchal permission process for the research and
ethical principle are described well in the method part and modified according to
the reviewer comment. 

Sampling techniques and sample size were not described. 

Sample techniques and sample size are stated in the method section under the
subheading “Sample size and sampling” and highlighted with red color. 

Quality assurance measures were not stated.

In revised manuscript, the Quality assurance measures are stated in the method
section under the subheading “Quality assurance measures”

Hospital and health center were used interchangeably (but not the
same)

The authors are grateful to the reviewer for arising this point. The authors are
agreed with reviewer that Hospital and health center are not the same and should not
be used interchangeably, thus the word “Hospital” is used in the entire manuscript
and highlighted with red color. 

Interpretation of the laboratory test for both viruses was not
stated!

We are grateful to the reviewer for this comment. The laboratory tests (ICT and
ELISA) were described in detail in the method section under subheadings “HBsAg and
HCV-Ab detection through ICT” and “HBsAg and HCV-Ab detection through ELISA”. The
interpretation of the test results are stated under the theses subheadings.

Data analysis steps were not clear to understand how it was performed

The authors are thankful to the reviewer for this comment. Data analysis statement
was described in detail in method section under subheading “Statistical analysis”
which can clearly be understood by the readers how the data was analyzed. 

Result

Result part was not elaborated well (e.g by describing the
socio-demographic factors except in the table)

We are obliged to reviewer for this comment. The result section is sub-headed in
“Socio demographic characteristics”. The socio-demographic characteristics are
described in detail under result section which explains all the results stated in
the tables. 

Prevalence of HBV and HCV, the socio-demographic factors and associated
factors shall be stated separately under different subtitles as the objective.

The authors are agreed with the reviewer comment. Prevalence of HBV and HCV, Socio
demographic characteristics and associated risk factors are stated separately under
different subheadings and highlighted with red color. 

In addition to table, if graphs are used to show the result, it will be
good to easily understand the result in %.

We appreciate the reviewer for this suggestion. In addition to tables, one graph was
created for showing the prevalence results in percentages and uploaded as a separate
file in the online Editorial Manager system according to reviewer comment.

There must be an explanation for why “Dental extraction, Receiving blood
transfusion, Educational level and Residential status associated with HBsAg
positivity and surgical procedure, maternal age for HCV Ab positivity” 

The authors are obliged to reviewer for this suggestion. Following the reviewer
suggestions, the Dental extraction, Receiving blood transfusion, Educational level
and Residential status associated with HBsAg positivity and surgical procedure,
maternal age for HCV Ab positivity” are explained in result section under the
subheadings “Socio demographic characteristics “, “Associated risk factors” and
“Prevalence of HBV and HCV Infections” and highlighted with red color.

Under discussion:

 Strength and Limitation of the study shall be specified

We are grateful to the reviewer for this comment. Strength and limitations of the
study are specified in the discussion section under the subheading “Study
limitations” and highlighted with red color.

Under Conclusion

 The result was not interpreted well whether it was high or low according to WHO
guidelines for HBV and HCV (whether it falls in the endemic interval or hyper
endemic interval and only risk factors were stated in line with the result.

The authors appreciate the reviewer for this constructive suggestion. Some new
sentences are added in the conclusion section which interpreted the results whether
the prevalence of HBV and HCV is high or low according to WHO guideline.

---

## [Decision Letter · Decision Letter 1]

25 Jun 2021

PONE-D-21-11040R1

Seroepidemiology and Associated Risk Factors of Hepatitis B and C Virus Infections
among Pregnant Women Attending Maternity Wards at Two Hospitals in Swabi, Khyber
Pakhtunkhwa, Pakistan

PLOS ONE

Dear Dr. Israr,

Thank you for submitting your manuscript to PLOS ONE. After careful consideration, we
feel that it has merit but does not fully meet PLOS ONE’s publication criteria as it
currently stands. Therefore, we invite you to submit a revised version of the
manuscript that addresses few the points raised during the review process.

If you would like to make changes to your financial disclosure, please include your
updated statement in your cover letter. Guidelines for resubmitting your figure
files are available below the reviewer comments at the end of this letter.

We look forward to receiving your revised manuscript.

Kind regards,

Isabelle Chemin, PhD

Academic Editor

PLOS ONE

Journal Requirements:

Reviewers' comments:

Reviewer's Responses to Questions

**Comments to the Author**

1. If the authors have adequately addressed your comments raised in a previous round
of review and you feel that this manuscript is now acceptable for publication, you
may indicate that here to bypass the “Comments to the Author” section, enter your
conflict of interest statement in the “Confidential to Editor” section, and submit
your "Accept" recommendation.

Reviewer #1: (No Response)

2. Is the manuscript technically sound, and do the data
support the conclusions?

Reviewer #1: Yes

3. Has the statistical analysis been performed
appropriately and rigorously? 

Reviewer #1: Yes

4. Have the authors made all data underlying the
findings in their manuscript fully available?

Reviewer #1: Yes

5. Is the manuscript presented in an intelligible
fashion and written in standard English?

Reviewer #1: Yes

6. Review Comments to the Author

Reviewer #1: Please see attachment to decision letter for review comments from
Reviewer #1

7. PLOS authors have the option to publish the peer
review history of their article (what does this mean?). If published, this will
include your full peer review and any attached files.

If you choose “no”, your identity will remain anonymous but your review may still be
made public.

**Do you want your identity to be public for this peer review?** For
information about this choice, including consent withdrawal, please see our
Privacy Policy.

Reviewer #1: No

#1 Minor correction for the author (1).docx
---

## [Author Response · Author response to Decision Letter 1]

26 Jun 2021

Responses to Reviewer Comments:

Comments were incorporated by the authors but I do have the following minor comments
to be considered! Suggestions were proposed in green colors as indicated below! 

Abstract part: Methodology 10 ml of blood was collected from each participated mother
…rather put as “From a total of 375 pregnant women selected using systematic random
sampling from both hospitals, 10ml of blood samples were collected…….

Response:

We are grateful to the reviewer for this suggestion. The above suggested sentence was
added in the revised version and highlighted with red color in track changes 

Under Method parts

Source and study population All pregnant women attending maternity wards for
antenatal care health facilities at Bacha Khan Medical Complex (BKMC) Shahmansoor
and District Head Quarter (DHQ) Hospital Swabi from surrounding urban and rural
areas of district Swabi during July 2019 -January 2020 were the source and study
population: Source population and study population is not the same 

Source populations were e all pregnant women attending hospital maternity wards for
antenatal care at Bacha Khan Medical Complex (BKMC) Shahmansoor and District Head
Quarter (DHQ) Hospital Swabi from surrounding urban and rural areas of district
Swabi 

Study population were all pregnant women attending hospital maternity wards for
antenatal care at Bacha Khan Medical Complex (BKMC) Shahmansoor and District Head
Quarter (DHQ) Hospital Swabi from surrounding urban and rural areas of district
Swabi during July 2019 -January 2020.

Response:

Thanks to arise this point. Yes, we are agree with reviewer that the source
population and study population are not the same and can be written in separate. In
order to follow the reviewer suggestion, the source population and study population
were added in method section under separate headings and highlighted with red color
in revised version with track changes.

Sample size and sampling ……better if it will be written as “Sample size and sampling
technique/procedure” The two hospitals were selected conveniently!

Response:

Thanks for this minor comment. The above sentence suggested by the reviewer was added
in the revised version and highlighted with red color in track changes. 

Sample collection

 10ml of blood samples were………………….better to write as “Ten milliliter of blood
samples were” starting with number is not recommended

Response: 

The authors are grateful to the reviewer for this minor comment. The above suggested
sentence was added in the revised version and highlighted with red color in track
changes.

Ethical endorsement

Management of positive tested participants not addressed? (Example: those
participants with positive test were given medication free of charge OR their result
was communicated with respective physician for further treatment)

Response: 

We are obliged to the reviewer for comment. The management of HBsAg and HCV-Ab
positive tested participant was incorporated with respective physician for further
treatment and addressed in the revised version under the ethical endorsement and
highlighted with red color in track changes.

Note: The whole manuscript was read for any other minor mistakes and corrected
accordingly.

---

## [Decision Letter · Decision Letter 2]

12 Jul 2021

Seroepidemiology and Associated Risk Factors of Hepatitis B and C Virus Infections
among Pregnant Women Attending Maternity Wards at Two Hospitals in Swabi, Khyber
Pakhtunkhwa, Pakistan

PONE-D-21-11040R2

Dear Dr. Israr,

We’re pleased to inform you that your manuscript has been judged scientifically
suitable for publication and will be formally accepted for publication once it meets
all outstanding technical requirements.

Kind regards,

Isabelle Chemin, PhD

Academic Editor

PLOS ONE

Additional Editor Comments (optional):

Reviewers' comments:

Reviewer's Responses to Questions

**Comments to the Author**

1. If the authors have adequately addressed your comments raised in a previous round
of review and you feel that this manuscript is now acceptable for publication, you
may indicate that here to bypass the “Comments to the Author” section, enter your
conflict of interest statement in the “Confidential to Editor” section, and submit
your "Accept" recommendation.

Reviewer #1: All comments have been addressed

2. Is the manuscript technically sound, and do the data
support the conclusions?

Reviewer #1: (No Response)

3. Has the statistical analysis been performed
appropriately and rigorously? 

Reviewer #1: (No Response)

4. Have the authors made all data underlying the
findings in their manuscript fully available?

Reviewer #1: (No Response)

5. Is the manuscript presented in an intelligible
fashion and written in standard English?

Reviewer #1: (No Response)

6. Review Comments to the Author

Reviewer #1: (No Response)

7. PLOS authors have the option to publish the peer
review history of their article (what does this mean?). If published, this will
include your full peer review and any attached files.

If you choose “no”, your identity will remain anonymous but your review may still be
made public.

**Do you want your identity to be public for this peer review?** For
information about this choice, including consent withdrawal, please see our
Privacy Policy.

Reviewer #1: No

---

## [Editor Report · Acceptance letter]

12 Aug 2021

PONE-D-21-11040R2 

Seroepidemiology and Associated Risk Factors of Hepatitis B and C Virus Infections
among Pregnant Women Attending Maternity Wards at Two Hospitals in Swabi, Khyber
Pakhtunkhwa, Pakistan 

Dear Dr. Israr:

I'm pleased to inform you that your manuscript has been deemed suitable for
publication in PLOS ONE. Congratulations! Your manuscript is now with our production
department. 

Kind regards, 

on behalf of

Mrs Isabelle Chemin 

Academic Editor

PLOS ONE